## COMMENT

# Increasing disability inclusion through self-relevant research

Kathleen R. Bogart [1]✉

Although often stigmatized in mainstream psychology, self-relevant research offers many benefits including increasing the presence of underrepresented researchers and promoting more valid and representative research. Psychology should de-stigmatize and leverage this approach.

In mainstream psychology, there is a stigma against conducting "me-search" or self-relevant research about one's own identity or experience[1]. This bias overshadows the fact that self-relevant research is common and can be a strength, especially by increasing inclusion of underrepresented minorities like people with disabilities. At 26% of the adult U.S. population[2], disability is the largest minority group and perhaps the only minority group you can join at any time. However, disabled researchers—and consequently, I would argue—disability research, is severely underrepresented in our field. In a report examining the years of 2006–2012, only 2% of faculty and 3% of students in APA-accredited programs reported a disability[3]. More recently, the National Science Foundation reported that only 3% of STEM workers had a disability in 2021[4].

Self-relevant research is widespread in the field of psychology, yet we neglect to see the most prevalent kind as such. Our field has long been criticized for sampling primarily Western, educated, industrialized, rich, and democratic (WEIRD) participants[5]. Interestingly, there is so little attention paid to disability in psychology that its absence has not even been noted in the acronym. I might propose adding a new letter, WEIRDA, for "abled." Unsurprisingly, WEIRDA samples match the demographics of the academics doing the research and the editorial teams of our top journals[6]. Thus, one might argue that WEIRDA samples are also self-relevant research.

When people in the majority group study the majority group, their research is perceived as universal and objective; when minority group members study experiences of their own group, it may be perceived as subjective and biased. This is because dominant identities, e.g. whiteness, abledness, etc., are perceived as the default or neutral[7]. In contrast, research on identities and experiences that are outside the dominant group is perceived as less important or relevant; it is seen as niche and relegated to "specialty journals." Thus, self-relevant research on minorities is more likely to be classified as such compared to self-relevant research on majority samples.

In a study of clinical, counseling, and school psychologists and students, more than half of participants reported engaging in self-relevant research[1]. Compared to majority group members, minorities were more likely to report conducting self-relevant work. This study also found that self-relevant researchers were rated as more biased and having poorer judgment compared to researchers who did not conduct research relevant to themselves. Participants who reported not engaging in self-relevant research made more stigmatizing judgements of self-relevant research than participants who reported having conducted self-relevant research. Just as majority group members may be more likely to examine their own groups' experiences, minoritized people may be more willing to examine experiences relevant to their group, but, as discussed above, minorities are more likely to be recognized as doing so. Stigma against self-relevant research serves to further marginalize the already marginalized, like people with disabilities.

Another reason self-relevant research is more common than it appears is that not all minority identities are visible or apparent. When a person has an invisible minoritized identity (e.g. a mental health condition or a chronic pain disorder), perceivers assume the default—as discussed

[1] School of Psychological Science, Oregon State University, Corvallis, OR, USA. ✉email: kathleen.bogart@oregonstate.edu

above—that the person is a majority group member. The distinction between visible and invisible identity affects the salience of self-relevant research, or whether it is identified as self-relevant research at all. As a visibly disabled researcher, anyone who views my headshot or listens to me speak at a conference can make the connection between my identity and my disability research. This experience is shared by many racial and ethnic minorities and some sexual and gender minorities whose identities are apparent. On the other hand, self-relevant researchers with non-apparent identities like mental health conditions would not be identified as such unless disclosed.

## A Personal Example

As a case study of the benefits and challenges involved with self-relevant research, I'll describe some of my own experiences as a self-relevant disability researcher. I was born with a rare neurological disorder, Moebius syndrome, which results in facial paralysis. Communicating in an unusual way made me fascinated with social interaction, drawing me to psychology. As an undergraduate, I set about to do a term paper on Moebius syndrome. However, I was frustrated to discover that there were only a handful of studies on the topic in psychology. The few extant studies examined Moebius syndrome in the service of better understanding "normal" processes. Critics have noted that majority group members researching minorities often take a deficit approach[6], with a goal of understanding dominant groups rather than understanding and improving the daily lives of minorities.

Recognizing the need to fill the gap in research on quality of life among people with Moebius syndrome and other conditions involving facial paralysis, I realized I had the unique motivation and insight to build this field. While pursuing graduate work, I found it challenging to find mentors. Psychology graduate programs follow an apprenticeship model, which hinges on finding a mentor who is an expert in your chosen field of study. The lack of psychologists studying facial paralysis combined with the paucity of role models with disabilities were barriers to finding a suitable graduate program. Eventually, I found excellent mentor-allies with expertize in broader areas that could be related to facial paralysis. At the start of my graduate work, I also connected with Moebius syndrome and facial paralysis communities for the first time. These connections were invaluable, providing me with an understanding beyond my own experiences of the issues facing this heterogeneous group of people.

I have since conducted some of the largest and most comprehensive psychosocial studies of people with facial paralysis. I have been fortunate to have had mainly positive experiences in academia, with a few exceptions. For instance, a few years ago, when I submitted a qualitative study of individuals with facial differences, I included a positionality statement noting that certain research team members had disabilities and facial differences. The statement discussed the challenges and benefits our insider identities presented and the steps we took to maximize rigor. We received mixed feedback on our positionality statement; one reviewer criticized the inclusion of researchers with disabilities as eliciting bias, while another commended the "strong positionality statement." The paper was rejected, but we ultimately found a home for it in a different journal.

## Benefits of self-relevant disability research

As exemplified in my experience, self-relevant disability researchers may be motivated to fill research gaps and make discoveries that otherwise would not be explored. Shared identity between researchers and participants builds trust and engagement, especially in marginalized populations that may mistrust science due to previous harms. Insider knowledge may promote more valid and representative research questions, study designs, sampling, interpretation, and implementation of findings. Thus, research is more likely to directly benefit the community.

## Challenges of self-relevant disability research

The most common argument against self-relevant research is that it interferes with objectivity[1]. Critics may argue that a self-relevant researcher may overweight their own perspective at the expense of others when conducting research. It might also be argued that self-relevant researchers could conduct research that places their ingroup in a more favorable light than the outgroup.

Of practical concern is that a self-relevant researcher may have preexisting relationships with participants and community organizations. If a participant knows a researcher, the participant may feel social pressure to participate or to respond in ways that the researcher might approve of. Similarly, researchers may be involved with community organizations that may be recruitment sites or even the subject of study. This can create role duality, where the researcher feels pressure to maintain a good relationship with the organization, while the research process may reveal participant concerns or complaints about the organization. Fortunately, there are ways to address these issues, which I will discuss in the next section.

## Maximizing the cost-benefit ratio of self-relevant research

**Considering positionality**. Positionality statements have been an important tradition in qualitative research. This recognizes that researchers come in with their own lived experiences, identities, and perspectives that shape the way they conduct research. Positionality statements are a transparent acknowledgement of the ways in which one's own perspective as a researcher has influenced one's work. Although traditionally, quantitative psychology has been less open to acknowledging positionality, there is a growing movement to include positionality statements in quantitative psychology research as well[6]. When majority group members engage in positionality statements, it may prompt reflection and encourage them to conduct more inclusive team science[6].

Those with invisible identities can choose not to disclose the self-relevance of their research, while those with visible or apparent identities do not have this privilege. Thus, it should be noted that positionality statements put people with invisible identities in a position where they may feel expected to disclose a self-relevant identity but do not feel safe doing so. Fear of disclosing due to stigma speaks to a larger problem in our field as a whole that must change, but in the meantime, it is important to consider this paradox.

**Team science**. Psychology research frequently takes a team science approach, which offers many benefits including diverse perspectives and skills. Many of the arguments against self-relevant research can be addressed with team science. A team may include several people with self-relevant experience, or a mixture of people with and without self-relevant experience. Steps must be taken to ensure that expectations of roles are clear and all perspectives of the team are valued to avoid tokenistic or inequitable inclusion of minorities.

**Participatory research**. Organizations and funders are now calling for community-based participatory research[8]. This approach asserts that community stakeholders with lived experience should co-produce science with researchers. Best practice guides for this type of research focus on decentering power structures, building knowledge, skills, and trust, and co-

creation at every level, in the service of developing research that solves the self-identified problems of communities. The challenge of role duality noted above can be addressed by having early transparent conversations about responsibilities and potential conflicts, and ensuring team members who have less role duality are also involved.

## Conclusion

As our field moves closer to fully open science, we need transparency around diversity. Psychology's WEIRDA self-relevant research will continue until we diversify our education, faculty, peer reviewers, and editorial boards. Destigmatizing disability self-relevant research will increase recruitment and retention of diverse researchers, further enhancing the science that is produced in our field.

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

## Author contributions

K.R.B. was responsible for all aspects of this manuscript.

## Competing interests

The author declares no competing interests.
