## [Peer Review File · Communications Psychology]

8th Dec 23

Dear Dr Bogart, Dear Kathleen,

Your manuscript titled "Nothing about us without us: Increasing disability inclusion through self-relevant research" has now been seen by our reviewers, whose comments appear below. In light of their advice I am delighted to say that we are happy, in principle, to publish a suitably revised version in Communications Psychology under the open access CC BY license (Creative Commons Attribution v4.0 International License).

We therefore invite you to revise your paper one last time to address the remaining concerns of our reviewers and a list of editorial requests. I have attached a marked up version of the manuscript to help guide the final edits. At the same time we ask that you edit your manuscript to comply with our format requirements and to maximise the accessibility and therefore the impact of your work.

EDITORIAL REQUESTS:

SUBMISSION INFORMATION:

In order to accept your paper, we require the files listed at the end of the Editorial Requests Table; the list of required files is also available at <https://www.nature.com/documents/commsj-file-checklist.pdf>. Please upload the revised comment as a word file (.docx).

OPEN ACCESS:

Communications Psychology is a fully open access journal. Articles are made freely accessible on publication under a [CC BY license](http://creativecommons.org/licenses/by/4.0) (Creative Commons Attribution 4.0 International License). This license allows maximum dissemination and re-use of open access materials and is preferred by many research funding bodies.

For further information about article processing charges, open access funding, and advice and support from Nature Research, please visit <https://www.nature.com/commspsychol/article-processing-charges>

At acceptance, you will be provided with instructions for completing this CC BY license on behalf of all authors. This grants us the necessary permissions to publish your paper. Additionally, you will be asked to declare that all required third party permissions have been obtained.

* TRANSPARENT PEER REVIEW: Communications Psychology uses a transparent peer review system. On author request, confidential information and data can be removed from the published reviewer reports and rebuttal letters prior to publication. If you are concerned about the release of confidential data, please let us know specifically what information you would like to have removed. Please note that we cannot incorporate redactions for any other reasons

[link redacted]

Best regards,

Jennifer

Jennifer Bellingtier, PhD
Senior Editor
Communications Psychology

REVIEWERS' EXPERTISE:

Reviewer #1 Diversity, Inclusion
Reviewer #2 Prejudices, stereotypes
Reviewer #3 Stigma

REVIEWERS' COMMENTS:

Reviewer #1 (Remarks to the Author):

In this review, the authors tackle the important topic of stigma associated with self-relevant research, highlighting both benefits and challenges to conducting such work. This commentary makes a valuable contribution to the growing body of work on diversity and equity as it examines an overlooked aspect of disability inclusion.

This is a hugely important topic, and the author has presented their reflections in a thought-provoking manner balancing both academic rigor with their personal experiences. In my review, I highlight one major and one minor issue that would further strengthen their arguments. I hope that this will allow the authors to improve an interesting manuscript.

Major:

Operationalization of the WEIRD-A label: Given that destigmatizing disability self-relevant research

forms the core argument for this submission, I find the combination of WEIRD and A as slightly detracting from the author's argument. I believe the overall message of the paper is very powerful, but I would urge the authors to reconsider their arguments moving beyond the binary conceptions of the WEIRD label. As most research even in Western contexts ignores ethnically and socio-economically diverse groups, and the WEIRD inadvertently label lumps social groups together, the addition of the term 'abled' can equally be perceived as part of a backronym. Please see Clancy and David (2019)'s review on Soylent Is People, and WEIRD Is White: Biological Anthropology, Whiteness, and the Limits of the WEIRD published in Annual Review of Anthropology for more explanation. I understand if the authors wish to use the WEIRD for the sake of rhetorical discussion. However, I find the statistic of "2% of faculty and 3% of students in APA-accredited programs reporting a disability (Andrews & Lund, 2015) as truly compelling which could benefit from further discussion.

The author could go into the consequences of the significantly low representation, addressing its impact not only on diverse scientific perspectives but also on career progression, including securing grant funding, fellowships, and future opportunities (if they see this as a fitting direction). The pressing need for disability inclusion within both the scientific community and the scientific workforce underscores the importance of addressing this issue. In revisiting the WEIRD-A comparison with researcher demographics, the WEIRD acronym may inadvertently suggest that researchers from non-Western backgrounds are inherently less educated or poor, which may not accurately reflect reality. Here the authors may find it beneficial to articulate the challenges and limitations inherent in labeling individuals as WEIRD. Furthermore, the exploration of who shapes and designs research leads to a broader discussion on the contributions of scientist diversity for new discoveries (see Medin, D. L. & Bang, M., "Who's Asking? Native Science, Western Science, and Science Education," MIT Press, 2014).

Minor:

The section on maximizing the cost-benefit ratio of self-relevant research provides a clear set of recommendations. The sub-section on team science made me curious whether the author believes team science projects dissolve the purpose of self-relevant research or advances it? Some clarification or discussions on equitable collaborations with an emphasis on researcher diversity might be helpful.

Overall, I commend the authors for picking such an important topic and addressing the lack of disability self-relevant research in our field.

Signed,^[1]_[SEP]
Sakshi Ghai

Reviewer #2 (Remarks to the Author):

This manuscript deals with the perception and importance of self-relevant research, notably in the field of disability. The main claim is that the stigma attached to self-relevant research should be lifted, notably because self-relevant research has many advantages that are relevant for the society in general (e.g., increasing the inclusion of underrepresented minorities) as well as for research in particular (e.g., improving the validity of research questions as well as representativeness).

These claims are novel when focusing on disability self-relevant research, even though some literature exists concerning self-relevant (or identity-relevant) research as a whole. Hence, I think this manuscript may be of interest to others in the community, but I would recommend taking into account a bit more previous research on this topic (see proposed references throughout this review).

Before developing my comments, I would like to underline the importance of this manuscript and its associated claim. I completely agree with the main claim concerning the importance of self-relevant research (concerning disability, as well as any other specific social group or minority). I am therefore really positive concerning the publication of this manuscript. Nevertheless, I think that this manuscript may not reach its goal in its current form. I will thus try to underline the points that I think may be strengthened. As you will see, my comments are more about form than substance, and I hope that you'll find them helpful.

General comments

- I believe that the paragraph on the benefits of self-relevant disability research should be more developed, as it is to me one of the main messages of this manuscript, or at least the most striking message to the scientific community. For this section, you may consider the following references:
 - o Amabile, T. M., & Hall, D. T. (2021). The undervalued power of self-relevant research: the case of researching retirement while retiring. *Academy of Management Perspectives*, 35(3).
<https://doi.org/10.5465/amp.2018.0083>
 - o Jones, E. B., & Bartunek, J. M. (2021). Too close or optimally positioned? The value of personally relevant research. *Academy of Management Perspectives*, 35(3).
<https://doi.org/10.5465/amp.2018.0009>
- On the whole, I found that the manuscript lacks some transitions to ease its reading. For example, in the introduction, you may consider adding at least a sentence to bridge the gap between the first and second paragraph. On the same vein, I found that the transition from the “Challenges of self-relevant disability research” section to the “Maximizing the cost-benefit ratio of self-relevant research” section was quite abrupt.
- I would suggest moving the section “A personal example” after the “Challenges [...]” section (and before the “Maximizing the cost-benefit [...]”) as I think it illustrates nicely the elements mentioned in the benefits and challenges sections.

More specific points

- In the introduction paragraph, I'd avoid mentioning first the common aspect of self-relevant research (i.e., [...] self-relevant research is common and can be [...]) and would focus on the strengths attached to this type of research (inclusion of people from a social minority, as well as improved validity of research questions and representativeness), that will be mentioned later in more depth.
- You underline that 26% of the adult US population has a disability, maybe you should consider adding information about the global population (i.e., 1 billion people, 15% of the global population according to the UN)
- I don't really understand the relevance of the last sentence of this first paragraph (i.e., “It is likely that self-reports [...]”) for the claim. I totally agree with this statement of underreport, but I'm not quite sure that it provides any added value in this specific context.
- I think that the paragraph concerning the WEIRD(A) samples is really striking, you may consider adding more information concerning the underrepresentation of research on disability. For example,

if you search in the literature referenced on Google Scholar since 2010, you'll find 76500 results for the "gender" and "discrimination" keywords, 94100 results for "race or ethnicity" and "discrimination" and only 26800 for "disability" and "discrimination" (on November 17th). That's not a really scientific point, but I find it quite meaningful. Maybe underline that this trend (underrepresentation) follows the general trend in society (e.g., 3.1% of characters with disability in TV shows in 2019, <https://www.respectability.org/2019/11/glaad-tv-report-2019/>; less than 1% in children's television programs, <https://seejane.org/research-informs-empowers/see-jane-2019/>)

- I'm not quite convinced by the third paragraph concerning the fact that self-relevant research is common and that all research is developed to understand something that people have observed or experienced in its current form. Not that I deeply disagree with that point, but it may seem a central claim and has no support of any kind in its current form. I am afraid that this statement might provokes more of a resistance for the readers, and thus impact the processing of subsequent elements.

You may want to reframe this paragraph by relying on the final paragraph of this section (the one beginning with "In a study of clinical, counseling, [...]"), as it provides some kind of support to this statement by underlying the common aspect of self-relevant research. Then you could continue with the paragraph developing the claim that self-relevant research on minorities is perceived as such because it is seen as deviating from the norm.

I think that this paragraph is a really important one (maybe the more striking in my opinion), and it should really be processed by the readers. In fact, I think this section on the perceived deviation from the norm could be developed a little further to support the point more strongly.

Notably, in line with the sentence "Stigma against self-relevant research only serves to further marginalized the already marginalized, like people with disability", I think that you may develop a little more the fact that self-relevant research from minorities is more likely to be classified as such compared to self-relevant research on majority sample. Indeed, I think it gives fuel to maintain or even reinforce the social categorization associated with minority and hence the stigma attached to these minorities (notably stereotypes) and their exclusion from society in general.

I think that the treatment of self-relevant research from minority members by the scientific community reinforce stigma, notably by creating a labelling of this kind of research (while no label is associated with research made by majority, even when it's on self-relevant subjects), thus reinforcing separation between minority and majority researchers (cf. Link & Phelan's definition of stigmatization, 2003: "stigma exists when elements of labeling, stereotyping, separation, status loss, and discrimination occur together in a power situation that allows them", p.377). I'm not sure how to include this point in the manuscript (and if it's necessary), but I wanted to share this idea.

You may also consider these references concerning perception and stigma of self-relevant research:

- o Rios, K., Roth, Z. C., & Langston, J. A. (pre-print). Academics' stereotypes about identity-related research: The role of intellectual humility. Retrieved from <https://psyarxiv.com/58y4k/download?format=pdf>

- o Devendort, A. R. (2020). Is me-search a kiss of death in mental health research? *Psychological Services*, 19(1), 49-54. <https://doi.org/10.1037/ser0000507>

- I am not sure that the paragraph concerning the visibility of the disability is relevant in its current form, even if I understand the claim here. You may consider including this idea more in line with the previous paragraph, and the idea that the attached stigma may not be avoided in the case of visible disability.
- Concerning participatory research, you may underline that we have no evidence (even scarce) concerning the benefits of this type of research, notably at the research question elaboration phase (see for example Bush et al., 2017)

Reviewer #3 (Remarks to the Author):

This piece presents valuable insight into the experiences of disabled researchers who study disability. The strengths and challenges of self-relevant research are discussed, and the paper addresses important issues with the discipline of psychology (e.g., issues with inclusivity, deficits approaches in the study of minoritized groups). I appreciated the powerful personal case study speaking to the benefits of conducting research from within the focal group. Sensible recommendations are provided to leverage the strengths of self-relevant research while offsetting potential costs (e.g., positionality, team science, co-production of science).

It would be beneficial to acknowledge the theoretical and empirical groundwork that has already been laid regarding this issue, albeit not in the domain of disability science.

Altenmüller, M. S., Lange, L. L., & Gollwitzer, M. (2021). When research is me-search: How researchers' motivation to pursue a topic affects laypeople's trust in science. *Plos One*, 16(7), e0253911. <https://doi.org/10.1371/journal.pone.0253911>

Veldhuis, C. B. (2022). Doubly marginalized: Addressing the minority stressors experienced by LGBTQ+ researchers who do LGBTQ+ research. *Health Education & Behavior*, 49(6), 960-974. <https://doi.org/10.1177/10901981221116795>

Thai, M., Lizzio-Wilson, M., & Selvanathan, H. P. (2021). Public perceptions of prejudice research: The double-edged sword faced by marginalized group researchers. *Journal of Experimental Social Psychology*, 96, 104181. <https://doi.org/10.1016/j.jesp.2021.104181>

Wallace, L. E., Craig, M. A., & Wegener, D. T. (2024). Biased, but expert: Trade-offs in how stigmatized versus non-stigmatized advocates are perceived and consequences for persuasion. *Journal of Experimental Social Psychology*, 110, 104519. <https://doi.org/10.1016/j.jesp.2023.104519>

I appreciated the consideration of both the benefits and challenges of positionality, and I would like to have seen this nuance extended to the discussion of team science and participatory research. For example, encouraging a team science approach may lead to tokenistic practices.